# The Decline and Fall of the Current Chemotherapy Paradigm in Soft Tissue Sarcoma

**DOI:** 10.3390/cancers17071203

**Published:** 2025-04-01

**Authors:** John Rieth, Varun Monga, Mohammed Milhem

**Affiliations:** 1Holden Comprehensive Cancer Center, University of Iowa Health Care, Iowa City, IA 52242, USA; mohammed-milhem@uiowa.edu; 2UCSF Helen Diller Family Comprehensive Cancer Center, San Francisco, CA 94158, USA; varun.monga@ucsf.edu

**Keywords:** sarcoma, chemotherapy, immunotherapy, adjuvant, neoadjuvant keyword, radiation, metastatic

## Abstract

Soft tissue sarcomas are diverse malignancies that are generally resistant to conventional chemotherapy. Little progress has been made with regard to the treatment of sarcoma, and chemotherapy, although largely ineffective, remains the standard of care. Further innovation in the study and use of novel systemic therapies are desperately needed, particularly with regard to targeted and immunotherapies. In particular, neoadjuvant clinical trials (treatment with systemic therapies prior to resection) provide an invaluable platform for the evaluation of novel treatments in patients and should be encouraged.

## 1. Introduction

Soft tissue sarcomas are a collection of diverse neoplasia thought to originate from mesenchymal progenitor cells. Over 70 subtypes of soft tissue sarcoma (STS) are known to exist, most recently classified by the WHO in 2020, with each histology possessing a unique natural history and sensitivity to different treatments [1,2]. STS is quite rare, constituting about 1–2% of all solid malignancies [1]. Frequently, the disease manifests as a localized tumor that can be cured with surgical resection, although the subsequent development of metastatic disease is common. Once metastatic, STS is generally considered incurable, with the rapid development of debilitation and death despite current treatments.

In his magnum opus, *The Structure of Scientific Revolutions* [3], American physicist and philosopher of science Thomas Kuhn describes the nature of the progression of scientific inquiry. In general, scientists operate under the common assumptions and premises of past scientific achievements, which he labels “normal science”. In this way, science slowly accumulates data that builds on and is guided by the current paradigm. Given the usefulness of certain paradigms for advancing scientific progress, scientists can frequently take these for granted, relying upon previously identified “truths” to advance their field of interest, building upon the work of others incrementally and no longer needing to attempt to build the scientific field anew from first principles.

Over time, however, inconsistencies arise that cannot be explained by the established paradigm. When a critical mass of problems with the current landscape arises, the consensus dissolves, resulting in “the proliferation of competing articulations, the willingness to try anything, the expression of explicit discontent, and the recourse to philosophy and to debate over fundamentals”. Thus, “extraordinary research” emerges by which the boundaries of science are expanded, novel theories are speculated, and experimentation is promoted.

In the field of sarcoma, chemotherapy has remained the dominant systemic treatment offered for the past 50 years in the adjuvant and neoadjuvant settings, as well as in the metastatic setting. Unfortunately, chemotherapy is largely ineffective in STS, with low response rates and few durable responses. Despite the development and approval of novel chemotherapeutic agents, overall survival (OS) has not significantly improved for patients with metastatic disease [4]. As more novel treatment options become available for many types of malignancies, an incremental change in the “chemotherapy alone” mindset for sarcoma is developing. Here, we discuss the current state of standard systemic treatment options for STS and propose suggestions to navigate the “paradigm shift” in sarcoma treatment, with a particular focus on adjuvant and neoadjuvant opportunities to identify novel treatments for STS.

## 2. Adjuvant and Neoadjuvant Cytotoxic Chemotherapy in Soft Tissue Sarcoma

The word “adjuvant” stems from the Latin word *adjuvare*, translated into English as “to help”. In 1957, the suggestion that chemotherapy may serve as an “adjunct” to surgery was identified in a murine model of breast carcinoma treated with 6-mercaptopurine, resulting in increased curing of the disease [5]. The authors hypothesized that chemotherapy could eliminate undetectable micrometastatic disease, resulting in a higher curative fraction. Subsequently, adjuvant chemotherapy was trialed in breast cancer [6], and further trials identified a significant increase in curative fraction [7]. Adjuvant chemotherapy was thus trialed in several other malignancies, including STS. In contrast, neoadjuvant therapy (from νέος meaning “new” in Greek) was subsequently pioneered as it was thought to reduce tumor burden at the primary site and improve surgical outcomes, in addition to assessing the treatment effect of the “neoadjuvant” modality [8].

Unlike the results in breast cancer, adjuvant and neoadjuvant cytotoxic chemotherapy has failed to provide robust evidence of an increase in the curative fraction in STS. This is likely due to several factors. First, sarcoma represents a highly heterogeneous collection of malignancies, which have varying biology, natural clinical history, and sensitivity to systemic treatment. Second, in general, STS is highly resistant to chemotherapy. In breast cancer, the response rates to chemotherapy are relatively high, with response rates to first-line chemotherapy frequently exceeding 50–60% [9]. By comparison, even with combination chemotherapy, the highest response rates in advanced STSs do not exceed 36%, and are often less, with combination doxorubicin and ifosfamide, one of the most frequently advocated adjuvant/neoadjuvant regimens, resulting in a 26% response rate [10,11]. Additional concerns with the tolerability of aggressive adjuvant and neoadjuvant chemotherapy have been raised, with toxicities that include cardiomyopathy, renal injury, development of secondary malignancies, and substantial reduction in quality of life. The toxicities are particularly relevant in neoadjuvant treatment, as toxicities may defer surgery and result in poorer outcomes. Furthermore, treating with chemotherapy in the adjuvant or neoadjuvant setting may promote the development of inducing resistant tumor clones, rendering treatment in the metastatic setting less effective for patients who unfortunately develop recurrent disease.

Numerous randomized adjuvant trials of chemotherapy in STS have been performed, with disappointing results. This is particularly the case in larger studies. A Gynecologic Oncology Group study involving 156 patients evaluated adjuvant doxorubicin 60 mg/m^2^ for the treatment of stage I and stage II uterine sarcoma, with no statistically significant differences in progression-free survival (PFS) or OS compared to untreated controls [12]. A randomized adjuvant trial by the Scandinavian Sarcoma Group randomized 240 patients with STS to doxorubicin 60 mg/m^2^ versus observation; again, no statistically significant differences in PFS or OS could be found [13]. A trial of CYVADIC (cyclophosphamide, vincristine, doxorubicin, dacarbazine) versus observation was performed by EORTC in patients with resected STS [14]. Although the reduced rate of local recurrence was identified, no differences in the development of metastatic disease or in OS were reported.

The current NCCN guidelines reflect the ISG-STS 1001 neoadjuvant study performed by the Italian (ISG), Spanish (GEIS), French (FSG), and Polish (PSG) sarcoma groups, published in 2020 by Gronchi et al. [15]. In this randomized phase III study, patients with high-risk grade 3 STS subtypes of the extremities or trunk wall, including high-grade myxofibrosarcoma, leiomyosarcoma, undifferentiated pleomorphic sarcoma, synovial sarcoma, malignant peripheral nerve sheath tumor, and undifferentiated pleomorphic sarcoma, were included. Patients were randomized to epirubicin with ifosfamide, or various other chemotherapy regimens varying between differing sarcoma subtypes, with an endpoint of identifying a reduced risk of relapse in the patients treated with the histotype-tailored regimens. Instead, the study was terminated early due to the underperformance of the histotype-tailored regimens, and the null hypothesis was reversed in favor of epirubicin and ifosfamide. Initially, epirubicin and ifosfamide were found to have superior disease-free survival [16], although a follow-up found this difference to not be statistically significant (*p* = 0.32) [15]. Furthermore, the randomization was not stratified using the histologic subtype, further complicating the interpretation of this study.

It has been hypothesized after the negative EORTC-STBSG 62931 adjuvant chemotherapy study [17] that perhaps there is a benefit that may still be obtained with adjuvant or neoadjuvant cytotoxic chemotherapy for patients with “high risk” STSs [18], but such hypotheses have yet to be evaluated in the setting of a randomized clinical trial against untreated controls. With the currently available data, neoadjuvant chemotherapy may be considered for patients with borderline resectable disease, where the chance of response to treatment may improve the chances of a negative margin operation or decrease the morbidity of a surgical operation, but a further clinical trial assessing adjuvant and neoadjuvant chemotherapy versus a placebo should be conducted prior to declaring this practice the standard of care.

## 3. Cytotoxic Chemotherapy in the Treatment of Unresectable or Metastatic Soft Tissue Sarcoma

Similar to adjuvant therapy, cytotoxic chemotherapy has demonstrated limited benefit in the management of most STSs. Doxorubicin has been seen as the cornerstone of sarcoma therapy for over fifty years, with the first data with regard to its use in STS dating back to the 1970s [19]. Multiple studies have been conducted comparing single-agent doxorubicin to multiple other different chemotherapeutic agents in the first-line setting for STS, including VAC [20], epirubicin [21,22], doxorubicin with dacarbazine [23,24,25], CYVADIC [26], docetaxel [27], doxorubicin and ifosfamide [10,26,28], pegylated liposomal doxorubicin [29], ifosfamide [30], conatumumab [31], brostallicin [32], trabectedin [33], aldoxorubicin [34], doxorubicin and palifosfamide [35], doxorubicin and trabectedin [36], doxorubicin and olaratumumab [37,38], doxorubicin and evofosfamide [39], and gemcitabine and docetaxel [40] Appendix A. Despite the diversity of agents used in these clinical trials, few trials have demonstrated the benefit with regard to the OS of an alternative chemotherapy regimen to doxorubicin, and likewise, few trials demonstrated the benefit of doxorubicin over the comparison chemotherapy arm, with the possible exception of the combination of trabectedin and doxorubicin for leiomyosarcoma [11]. Given the previously mentioned first-line studies, with essentially no differences in OS between the control and experimental groups, there is no clear first-line cytotoxic therapy in STS.

Unfortunately, chemotherapy in metastatic STS is curative in only an extremely small number of cases where tumor regression may permit surgical excision. Studies have been conducted to attempt to identify molecular signatures that may correspond to chemosensitivity, but such techniques have yet to be employed in clinical practice [41]. Given the marginal improvement in OS of STS despite novel chemotherapeutics, it is difficult to believe that any future development of non-targeted cytotoxic chemotherapy will result in durable outcomes. Furthermore, most chemotherapy regimens for STS are difficult to tolerate, resulting in severe toxicities for many patients, substantially reducing quality of life.

## 4. Quo Vadis, Sarcoma Oncologist?

Chemotherapy has largely been disappointing for the treatment of STS, with relatively few durable responses and essentially no cures for patients with metastatic disease. From our perspective, several ideas may be implemented to improve care for patients with STS.

### 4.1. Exploration of Therapies with Alternative Mechanisms of Action

Of all the possible outcomes of relevance within the field of oncology, patient-centered outcomes are the most important. Patients generally have two goals of treatment: an increase in longevity and improvement or maintenance of quality of life. Unfortunately, standard chemotherapy treatments for sarcoma are frequently ineffective and also substantially impair quality of life. Due to the disappointment of cytotoxic chemotherapy in sarcoma, novel treatments must be evaluated in STS. It may be wise to look towards our colleagues who treat other tumor subtypes, including those who treat lung cancer, melanoma, breast cancer, renal cell carcinoma, and hematological malignancies for suggestions of alternative therapies. Novel drug targets and therapies should also be evaluated in different histological subtypes in pre-clinical studies.

Immunotherapy has been used to treat sarcoma prior to chemotherapy, starting with the pioneering work of Dr. Coley and his famous (or infamous) toxins [42]. Currently, preliminary success has been demonstrated in STS, which has been demonstrated by immunotherapies, including checkpoint inhibitors and, more recently, CAR-T-cell therapy. To date, most successes in checkpoint inhibitor therapy have occurred for undifferentiated pleomorphic sarcoma, dedifferentiated liposarcoma, and angiosarcoma, resulting in modest response rates for each, both in clinical trials and in the real-world setting [43,44]. The early results of a randomized phase II study evaluating cabozantinib combined with ipilimumab versus cabozantinib alone demonstrate an improved response rate and PFS with the immunotherapy arm, providing further early support for the study of checkpoint inhibitors in STS [45]. More interestingly, however, responses to checkpoint inhibitors are potentially durable, unlike responses to chemotherapy. The single-arm phase II SARC028 clinical trial identified several patients with undifferentiated pleomorphic sarcoma and liposarcoma with durable responses lasting for years [46]. An additional advantage of checkpoint inhibitor therapy is the tolerability of such treatment and that toxicities are generally easily managed, providing a good quality of life for patients treated with these agents [47]. Currently, checkpoint inhibitors may have limited benefits for metastatic STS, but there is clearly an opportunity to build on current observations to improve both the immediate responses and depth of response of these therapies in STS.

The durability of responses has also been identified in CAR-T-cell therapy as well for synovial sarcoma, with a recently published phase II trial finding durable responses with afamitresgene autoleucel, a CAR-T-cell product targeting MAGE-A4 [44]. Given the durability of responses to immune therapy, significant investment should be made to identify partner treatments that can increase the response rate of immunotherapy in sarcoma.

Except in the case of GIST, tyrosine kinase inhibitors and other molecular-targeted therapies have had limited success in STS. The only current tyrosine kinase inhibitor approved in the United States for most STS subtypes is pazopanib, originally evaluated in the PALLETTE trial versus placebo, with a marginal improvement in PFS but no OS benefit [48]. Pazopanib was found to be non-inferior to doxorubicin in elderly patients with STS [49]. Other tyrosine kinase inhibitors have demonstrated preliminary signs of efficacy, including cabozantinib [45] and lenvatinib [50]. Antibody–drug conjugates represent an interesting avenue of research that should be further explored, especially given the successes in HER-2-expressing cancers [51,52] and urothelial carcinoma [53]. Evaluations of metabolic therapies are also intriguing, with the ongoing ARGSARC trial evaluating arginine deprivation as a mechanism to augment the antitumoral effects of chemotherapy [54].

### 4.2. Novel Treatments in the Neoadjuvant Setting

Neoadjuvant clinical trials serve as a unique opportunity to evaluate novel treatment effects in STS. One of the current accepted standard treatments for localized STS is definitive radiation over 5–6 weeks, followed by surgical resection; neoadjuvant hypo-fractionated radiation is also under investigation [55]. This provides ample opportunity to design window-of-opportunity trials, using the time between the initiation of radiation treatment and surgical resection as a time to evaluate novel treatments for STS. Neoadjuvant window-of-opportunity trials offer invaluable information that is difficult to obtain in trials of metastatic sarcoma, as the treated sarcoma tumor specimen is resected after treatment. This is ideal for identifying the treatment effects of novel agents on the malignant cells, immune system, and the tumor microenvironment, giving us insight not only into response rates of various novel treatments but enhancing our knowledge of sarcoma biology as well.

The importance of innovation in the neoadjuvant space was demonstrated by SARC032 [56]. This bold randomized phase II trial evaluated the addition of pembrolizumab to concurrent neoadjuvant radiation followed by adjuvant pembrolizumab in patients with localized extremity undifferentiated pleomorphic sarcoma and liposarcoma. This demonstrated significant improvements in disease-free survival in this patient population, particularly in patients with a grade 3 disease. The success of this trial provides a foundation that future trials involving the immune system can build upon [57].

The sarcoma team at Case Western Reserve evaluated pazopanib when combined with radiation in STS, with the hypothesis that hypoxia due to the VEGF inhibition of pazopanib may augment radiation efficacy [58]. Interestingly, this combination results in a major pathological response (defined as a treatment effect greater than 90%) in 70% of patients, although with a small sample size. These promising, yet very preliminary findings provide an excellent rationale for the assessment of neoadjuvant pazopanib in a randomized clinical trial and should inspire other researchers to consider the effects of tyrosine kinase inhibitor-induced tumor hypoxia in treatment combination for STS.

Over the past several years, we have designed trials with the goal of invoking antitumoral immune responses and breaking immune tolerance. In the first trials, the oncolytic virus Talimogene laherparapvec (TVEC) was injected directly into localized high-risk STS with concurrent neoadjuvant radiation therapy [59,60]. Initially, 4 mL injections of intralesional TVEC were evaluated, as this dose had previously been approved for melanoma [61]. This trial demonstrated several interesting observations, with increased intratumoral lymphoplasmacytic infiltration compared to historical controls, and a few patients with metastatic disease developed antitumoral responses in metastatic lesions [60]. Given these findings, the injectable dose was escalated to 8 mL in a subsequent phase I trial [59]. A total of eight patients were treated, with subtypes including undifferentiated sarcoma, dedifferentiated liposarcoma, and myxoid round-cell liposarcoma. Laboratory analysis of the treated tumors demonstrated an increase in CD8^+^ T-cells compared to tumors treated with radiation alone, suggestive of immune activation. This is currently being further evaluated in an expanded study.

An additional phase IB/II neoadjuvant trial involved combining pharmacological ascorbate with neoadjuvant radiation [62]. Tissue from the resected treated tumors identified CD8^+^ T-cell infiltration that was correlated with increased tumor necrosis, with associated blood CD8^+^ T-cells also increased with continuation of treatment. The above findings with enhanced CD8^+^ T-cell presence in sarcomas treated with both TVEC and with pharmacological ascorbate both suggest that STS may be a more immunologically sensitive malignancy than first thought, and they also provide a rationale for combination therapies with oncolytic viral therapy, pharmacological ascorbate, and checkpoint inhibitors.

Despite the above-stated advantages for neoadjuvant window-of-opportunity trials, substantial changes remain. Sarcomas are rare and diverse, and it is likely that novel therapies may have different treatment effects between sarcoma STS subtypes. This necessitates that each intervention should be tested in multiple tumor types so that efficacy signals are not missed. Furthermore, alternatives to the commonly used endpoint of “necrosis” must be explored due to the variability of necrosis rates between STS subtypes and because necrosis is not generally reflective of outcomes including PFS and OS [63]. Future trials should explore alternative response metrics that may improve outcomes relevant to patients, including pathologic complete responses, radiologic responses such as RECIST criteria, molecular immune activation markers, or circulating tumor DNA dynamics.

## 5. Conclusions

Outcomes for STS have not improved substantially over the past several decades. Despite the approvals of several chemotherapy agents over the past decade, outcomes remain poor. It is clear that chemotherapy is largely ineffective at providing durable responses in most patients with STS, is highly toxic, and substantially decreases quality of life. More innovative experimentation with novel therapeutics is needed.

Novel immunotherapies and targeted therapies are being evaluated in current clinical trials, and more such clinical trials should be encouraged. In particular, we would like to emphasize the importance of neoadjuvant trials for soft tissue sarcoma, both as a means of furthering our understanding of sarcoma and the effects that antitumoral treatments have on sarcoma by an analysis of resected treated tumors. The delay between the start of radiation and surgical resection provides the ideal length of time for such window-of-opportunity trials. Only by challenging the current paradigm of chemotherapy in STS may substantially improved outcomes be obtained.

## Data Availability

No new data were created or analyzed in this study. Data sharing is not applicable to this article.

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
