# Peer review of "The Decline and Fall of the Current Chemotherapy Paradigm in Soft Tissue Sarcoma"

_cancers, 2025, doi:10.3390/cancers17071203_

Round 1

Reviewer 1 Report

Comments and Suggestions for Authors

The authors provide an excellent mini-review/overview of systemic treatments for STS including the limited efficacy of traditional chemotherapy as well as some newer promising results with immunotherapies and targeted therapies. The manuscript is well written and will be of interest to the sarcoma and oncology communities.

  • No major suggestions for revision.
  • Minor points:
    • It would be helpful if the authors included a few representative studies of checkpoint inhibitors in STS (section 4 lines 166-170).
    • A graphic, summary table, or timeline of treatments would be a useful addition. 

Author Response

The authors provide an excellent mini-review/overview of systemic treatments for STS including the limited efficacy of traditional chemotherapy as well as some newer promising results with immunotherapies and targeted therapies. The manuscript is well written and will be of interest to the sarcoma and oncology communities.

  • No major suggestions for revision.
  • Minor points:
    • It would be helpful if the authors included a few representative studies of checkpoint inhibitors in STS (section 4 lines 166-170).
    • A graphic, summary table, or timeline of treatments would be a useful addition. 

We would like to thank the reviewer for their kind suggestions. We have citations of examples of both clinical studies and real world data of the use of checkpoint inhibitors in soft tissue sarcoma.  The changes can be found in the fourth section in lines 184-203.

Reviewer 2 Report

Comments and Suggestions for Authors

The  authors Rieth and colleagues provide a commentary about the limited efficacy of current chemotherapy in soft tissue sarcoma.

The work is interesting and has focused on an hot topic.

The manuscript would benefit from the followings:

  1. Lates Who should be included: WHO Classification of Tumours Editorial Board. WHO Classification of Tumours of Soft Tissue and Bone, 5th ed.” Lyon, France: IARC Press; 2020
  2. A table resuming all the chemotherapy investigated in soft tissue sarcoma would would facilitate the readers of the manuscript
  3. In the Exploration of Therapies with Alternative Mechanisms of Action section the following manuscript should be referenced for proper discussions:

Unveiling the Genomic Basis of Chemosensitivity in Sarcomas of the Extremities: An Integrated Approach for an Unmet Clinical Need. Int J Mol Sci. 2023 Apr 8;24(8):6926. doi: 10.3390/ijms24086926. PMID: 37108089; PMCID: PMC10138892.

Feasibility and safety study of ultra-hypofractionated neoadjuvant radiotherapy to margins-at-risk in retroperitoneal sarcoma. Radiat Oncol J. 2025 Jan 15. doi: 10.3857/roj.2024.00297. Epub ahead of print. PMID: 39928964.

Heterogeneity in response to neoadjuvant radiotherapy between soft tissue sarcoma histotypes: associations between radiology and pathology findings. Eur Radiol. 2025 Mar;35(3):1337-1350. doi: 10.1007/s00330-024-11258-6. Epub 2024 Dec 19. PMID: 39699680.

Neoadjuvant immunotherapy in the evolving landscape of sarcoma treatment. J Immunother Cancer. 2024 Nov 24;12(11):e010074. doi: 10.1136/jitc-2024-010074. PMID: 39581705; PMCID: PMC11590830.

Molecular Determinants of Soft Tissue Sarcoma Immunity: Targets for Immune Intervention. Int J Mol Sci. 2021 Jul 13;22(14):7518. doi: 10.3390/ijms22147518. PMID: 34299136; PMCID: PMC8303572.

  1. Study limitations should be included
Comments on the Quality of English Language

The quality of the english is good

Author Response

  1. Lates Who should be included: WHO Classification of Tumours Editorial Board. WHO Classification of Tumours of Soft Tissue and Bone, 5th ed.” Lyon, France: IARC Press; 2020

We would like to thank the reviewer for their kind suggestions. To improve the manuscript, a citation to the WHO classification of Tumours was added to lines 36-37.

  1. A table resuming all the chemotherapy investigated in soft tissue sarcoma would would facilitate the readers of the manuscript

A table including prior anthracycline chemotherapy trials in soft tissue sarcoma was added.

  1. In the Exploration of Therapies with Alternative Mechanisms of Action section the following manuscript should be referenced for proper discussions:

The recommended citations were added to the manuscript.

  1. Study limitations should be included

The limitations of our proposals was added immediately prior to the conclusions in lines 273-283.  

Again, thank you for your kind words and helpful suggestions.

Reviewer 3 Report

Comments and Suggestions for Authors

General Comments

This reviewer would like to commend the authors for their timely and well-structured manuscript addressing the long-standing and widely acknowledged limitations of chemotherapy in soft tissue sarcoma (STS). The paper presents an important and much-needed critical evaluation of the current systemic treatment landscape for STS and advocates for a necessary shift towards novel therapeutic approaches. The historical framing using Kuhn’s paradigm shift theory is an intellectually stimulating angle, and the discussion of neoadjuvant clinical trials as a platform for innovation is highly relevant.

However, while the manuscript provides a strong and compelling argument, there are several key areas where improvements are necessary to ensure a scientifically robust and clinically meaningful discussion. Below, we outline specific points that should be addressed to strengthen the article.

Major Comments

  1. Defining Relevant Clinical Endpoints for Neoadjuvant Trials
  • The manuscript strongly advocates for neoadjuvant trials as a research opportunity but does not sufficiently address the issue of clinically meaningful endpoints beyond tumor necrosis.
  • As shown in multiple studies, tumor necrosis is a poor predictor of overall survival (OS), metastasis-free survival (MFS), and local recurrence (LR). Thus, while it may serve as a readout of treatment effect, it is not a surrogate for meaningful clinical benefit.
  • Recommendation: The authors should discuss alternative response metrics that could better predict patient-relevant outcomes, such as pathologic complete response (pCR), radiologic response (e.g., RECIST criteria), molecular immune activation markers, or circulating tumor DNA (ctDNA) dynamics.
  1. Immunotherapy and Multimodal Therapy – A Critical Reevaluation
  • The authors advocate for immunotherapy and targeted therapies in sarcoma but do not provide a critical discussion of their actual efficacy in STS.
  • Recent trials—including SARC032 (Kirsch et al., Lancet, Nov 2024)—demonstrated a statistically significant delta favoring pembrolizumab in a neoadjuvant setting. However, this difference was not clinically meaningful because:
    1. The control arm performed worse than historical controls, making the observed benefit questionable.
    2. The duration of therapy (12 months) is substantially longer than conventional systemic treatment, raising concerns about toxicity and patient burden.
    3. Considering toxicity, cost, and the modest benefit observed, the data do not support widespread adoption of this approach.
    4.  
  • Recommendation: The authors shall acknowledge that the overall impact of checkpoint inhibitors in STS remains marginal, and highlight that any systemic treatment must be evaluated in terms of a benefit-risk-cost efficiency framework rather than simply assuming new treatments are inherently better.
  1. Neoadjuvant vs. Adjuvant Therapy – A Lack of Differentiation
  • The manuscript does not clearly distinguish between neoadjuvant and adjuvant chemotherapy trials and their respective implications.
  • In clinical practice, adjuvant chemotherapy may be preferable in some cases, particularly when tumor shrinkage is not a primary objective.
  • Potential downsides of neoadjuvant chemotherapy should also be addressed, including:
    • The risk of inducing resistant clones if the neoadjuvant regimen is ineffective.
    • Potential delays in surgery due to lack of response or local and/or systemic toxicity.
  • Recommendation: A clearer comparison of neoadjuvant vs. adjuvant strategies should be made, discussing their relative advantages, risks, and indications in different sarcoma subtypes.
  1. Cost-Efficiency and Real-World Data: A Critical Omission
  • The absence of a cost-benefit analysis is a major weakness in the manuscript.
  • Immunotherapies, targeted therapies, and CAR-T treatments are significantly more expensive than chemotherapy, yet their benefits in STS remain marginal at best.
  • Real-world data (e.g., from international sarcoma registries) should be integrated to assess whether novel therapies improve long-term outcomes in a cost-effective manner.
  • Recommendation: The authors should include a dedicated discussion on:
    • Economic feasibility of new treatment approaches.
    • The importance of using real-world data to validate findings beyond clinical trial populations.
    • Value-based healthcare models as a framework for evaluating emerging therapies.
  1. Neoadjuvant Therapy is Not Necessarily “Underutilized” – But Needs Global Standardization
  • The manuscript suggests that neoadjuvant therapy is an "underutilized" strategy, but this is misleading.
  • Many sarcoma centers are already conducting neoadjuvant trials, but the true issue is fragmentation and lack of standardized international collaboration.
  • Recommendation: Instead of claiming underutilization, the manuscript should emphasize the need for global, harmonized clinical trial efforts to:
    • Define common inclusion criteria and outcome measures.
    • Standardize radiologic and histologic response metrics.
    • Pool data across multiple centers to increase statistical power.
  1. The Kuhn Paradigm Shift Theory: Is It the Right Analogy?
  • Kuhn’s Structure of Scientific Revolutions describes how a scientific discipline undergoes a fundamental shift when accumulated contradictions force the abandonment of a prevailing paradigm.
  • However, sarcoma treatment has already been shifting towards precision medicine and targeted therapy for years, making the use of this theory somewhat forced.
  • The authors present their argument as if we are on the brink of a dramatic revolution, but in reality, the transition is more incremental than revolutionary.
  • Recommendation: The authors should reconsider whether this is truly a paradigm shift or simply a gradual refinement of sarcoma treatment. A more balanced discussion would be appropriate.

This reviewer genuinely appreciates the authors’ significant contribution to this critical discussion in STS treatment. With these refinements, this paper could become an authoritative reference in shaping future sarcoma therapeutic strategies.

Author Response

  1. Defining Relevant Clinical Endpoints for Neoadjuvant Trials
  • The manuscript strongly advocates for neoadjuvant trials as a research opportunity but does not sufficiently address the issue of clinically meaningful endpoints beyond tumor necrosis.
  • As shown in multiple studies, tumor necrosis is a poor predictor of overall survival (OS), metastasis-free survival (MFS), and local recurrence (LR). Thus, while it may serve as a readout of treatment effect, it is not a surrogate for meaningful clinical benefit.
  • Recommendation: The authors should discuss alternative response metrics that could better predict patient-relevant outcomes, such as pathologic complete response (pCR), radiologic response (e.g., RECIST criteria), molecular immune activation markers, or circulating tumor DNA (ctDNA) dynamics.

We would like to thank the reviewer for their kind and detailed suggestions to improve the quality of our manuscript.  With regards to the comment referring to relevant clinical trial endpoints, we have added the recommendation that future neoadjuvant studies in sarcoma provide an alternative primary outcome to necrosis, which we recognize is a poor surrogate for outcomes that are relevant to patients.  We have addended the manuscript in lines 277-283 to address this.

  1. Immunotherapy and Multimodal Therapy – A Critical Reevaluation
  • The authors advocate for immunotherapy and targeted therapies in sarcoma but do not provide a critical discussion of their actual efficacy in STS.
  • Recent trials—including SARC032 (Kirsch et al., Lancet, Nov 2024)—demonstrated a statistically significant delta favoring pembrolizumab in a neoadjuvant setting. However, this difference was not clinically meaningful because:
    1. The control arm performed worse than historical controls, making the observed benefit questionable.
    2. The duration of therapy (12 months) is substantially longer than conventional systemic treatment, raising concerns about toxicity and patient burden.
    3. Considering toxicity, cost, and the modest benefit observed, the data do not support widespread adoption of this approach.
      Recommendation: The authors shall acknowledge that the overall impact of checkpoint inhibitors in STS remains marginal, and highlight that any systemic treatment must be evaluated in terms of a benefit-risk-cost efficiency framework rather than simply assuming new treatments are inherently better.
    4.  

With regards to the critical re-evaluation of immunotherapy, we agree with the reviewer on several points and keen observations. We understand that current benefit of checkpoint inhibitors in soft tissue sarcoma is modest at best, and this point was re-emphasized in the paragraph in checkpoint inhibitors in lines 187-203.  With regards to SARC032, we also recognize the concerns that were mentioned by the reviewer.  It is true that the control arm performed worse than historical controls, but this may be due to several factors, including the enrichment of grade 3 undifferentiated pleomorphic sarcoma in this study, which is known to have very poor outcomes in general.  We agree that the widespread adoption of checkpoint inhibitors is premature for STS, and thus we recommend further investigation by way of clinical trials in the final sections of the manuscript.

  1. Neoadjuvant vs. Adjuvant Therapy – A Lack of Differentiation
  • The manuscript does not clearly distinguish between neoadjuvant and adjuvant chemotherapy trials and their respective implications.
  • In clinical practice, adjuvant chemotherapy may be preferable in some cases, particularly when tumor shrinkage is not a primary objective.
  • Potential downsides of neoadjuvant chemotherapy should also be addressed, including:
    • The risk of inducing resistant clones if the neoadjuvant regimen is ineffective.
    • Potential delays in surgery due to lack of response or local and/or systemic toxicity.
  • Recommendation: A clearer comparison of neoadjuvant vs. adjuvant strategies should be made, discussing their relative advantages, risks, and indications in different sarcoma subtypes.

Additionally, we agree that a stronger distinction between neoadjuvant and adjuvant chemotherapy was needed, and we better defined adjuvant from neoadjuvant treatment in section 2, line 80-83.  We agree with the potential down sides of neoadjuvant therapy, and have added these downsides to the manuscript in lines 94-101.  We have added a sentence with regards to the potential induction of resistance in patients treated with neoadjuvant chemotherapy, which may render further treatment upon the development of metastatic disease inefficacious in lines 98-101.  With regards to discussing adjuvant/neoadjuvant therapy in different sarcoma subtypes, this us unfortunately beyond the scope of this review, given the focus on development of novel therapies.

  1. Cost-Efficiency and Real-World Data: A Critical Omission
  • The absence of a cost-benefit analysis is a major weakness in the manuscript.
  • Immunotherapies, targeted therapies, and CAR-T treatments are significantly more expensive than chemotherapy, yet their benefits in STS remain marginal at best.
  • Real-world data (e.g., from international sarcoma registries) should be integrated to assess whether novel therapies improve long-term outcomes in a cost-effective manner.
  • Recommendation: The authors should include a dedicated discussion on:
    • Economic feasibility of new treatment approaches.
    • The importance of using real-world data to validate findings beyond clinical trial populations.
    • Value-based healthcare models as a framework for evaluating emerging therapies.

In agreement with the reviewer, we also agree that cost efficiency is of critical importance in the development of new treatment strategies.  We recognize the high cost and checkpoint inhibitors, CAR-T treatments, targeted therapies and other novel treatments, particularly when compared to relatively inexpensive chemotherapy.  We also agree that more needs to be done to study real world outcomes in patients treated with all therapies, and newer therapies in particular, and that futures studies should be performed to assess access to novel treatments and to assess the value of such therapies.  Unfortunately, we must admit that this is beyond the scope of this review.

  1. Neoadjuvant Therapy is Not Necessarily “Underutilized” – But Needs Global Standardization
  • The manuscript suggests that neoadjuvant therapy is an "underutilized" strategy, but this is misleading.
  • Many sarcoma centers are already conducting neoadjuvant trials, but the true issue is fragmentation and lack of standardized international collaboration.
  • Recommendation: Instead of claiming underutilization, the manuscript should emphasize the need for global, harmonized clinical trial efforts to:
    • Define common inclusion criteria and outcome measures.
    • Standardize radiologic and histologic response metrics.
    • Pool data across multiple centers to increase statistical power.

With regards to the utilization of neoadjuvant therapy, we agree that neoadjuvant therapy is not necessarily “underutilized.”  We simply suggest that evaluating novel therapies in the neoadjuvant space as a “window of opportunity” as an ideal setting with which to evaluate new treatments, as mentioned in line 226-228.  We would also like to recognize that we agree with the need for global harmonized clinical trial efforts in the neoadjuvant space to clarify optimal treatment patterns for different subtypes of soft tissue sarcoma, but this is beyond the scope of this review.

  1. The Kuhn Paradigm Shift Theory: Is It the Right Analogy?
  • Kuhn’s Structure of Scientific Revolutions describes how a scientific discipline undergoes a fundamental shift when accumulated contradictions force the abandonment of a prevailing paradigm.
  • However, sarcoma treatment has already been shifting towards precision medicine and targeted therapy for years, making the use of this theory somewhat forced.
  • The authors present their argument as if we are on the brink of a dramatic revolution, but in reality, the transition is more incremental than revolutionary.
  • Recommendation: The authors should reconsider whether this is truly a paradigm shift or simply a gradual refinement of sarcoma treatment. A more balanced discussion would be appropriate.

This reviewer genuinely appreciates the authors’ significant contribution to this critical discussion in STS treatment. With these refinements, this paper could become an authoritative reference in shaping future sarcoma therapeutic strategies.

With the last suggestion, we also agree that we are incrementally changing from chemotherapy to alternative systemic treatment options.  We have altered the introduction to reflect this in lines 65-67.  We do, however, recognize that there is a substantial difference between the prior chemotherapy regimens that have been used for decades, and the newer treatments that have been tested in the past few years.  Even if this change has been relatively incremental, we feel that the Kuhn Paradigm shift is still applicable, given the incremental novelty of treatment strategies being proposed in this space.

Again, we very much appreciate the reviewers helpful suggestions and kind compliments.

Reviewer 4 Report

Comments and Suggestions for Authors

The Paper “The Decline and Fall of the Current Chemotherapy Paradigm in Soft Tissue Sarcoma” is an interesting commentary regarding the current knowledge in the therapy of soft tissue sarcoma. The authors have made a clear statement about the aim of the paper and have been clear regarding the presentation of the current data. Some minor adjustments could improve the paper.

Comments:

  1. In the section “Adjuvant and Neoadjuvant Cytotoxic Chemotherapy in Soft Tissue Sarcoma” as well as in “Cytotoxic Chemotherapy in the Treatment of Unresectable or Metastatic Soft Tissue Sarcoma” there is clear perspective regarding the efficacy of the therapy. However, there is a lack regarding the side effects of the therapy and how the patients tolerate the therapy that is very aggressive. Also, a few lines regarding the quality of life of the patients under all therapeutic choices should be put in perspective.
  2. In the part 4. under each subsection the side effects and the safety profile of each therapeutics mentioned should be extracted from the mentioned studies in order to point out also the limitations of the therapeutics. Throughout the history of medicines, most of the drugs that have been withdrawn from the market, had serious and/or life-threatening side effects. So mentioning the safety profile of the experimental therapies or drugs in undergoing clinical studies is mandatory.
  3. The most important statements regarding the side effects of the current therapy and the tolerability of the patients to the therapy should be added in the conclusion as well. This could be good starting point for further development of improved therapeutics given the limitations in efficiency of the therapeutic choices as well as the safety profile of the available therapy.

Author Response

Comment 1: In the section “Adjuvant and Neoadjuvant Cytotoxic Chemotherapy in Soft Tissue Sarcoma” as well as in “Cytotoxic Chemotherapy in the Treatment of Unresectable or Metastatic Soft Tissue Sarcoma” there is clear perspective regarding the efficacy of the therapy. However, there is a lack regarding the side effects of the therapy and how the patients tolerate the therapy that is very aggressive. Also, a few lines regarding the quality of life of the patients under all therapeutic choices should be put in perspective.

Response 1: We would like to thank the reviewer for their kind suggestions. We have elaborated on the poor tolerability of intensive chemotherapy regimens in STS in lines 94-98 and in lines 165-167.  We have also added a section on the improved tolerability of checkpoint inhibitors relative to chemotherapy in lines 197-200.

Comment 2: In the part 4. under each subsection the side effects and the safety profile of each therapeutics mentioned should be extracted from the mentioned studies in order to point out also the limitations of the therapeutics. Throughout the history of medicines, most of the drugs that have been withdrawn from the market, had serious and/or life-threatening side effects. So mentioning the safety profile of the experimental therapies or drugs in undergoing clinical studies is mandatory.

Response 2: We added a section on the improved tolerability of checkpoint inhibitors relative to chemotherapy in lines 197-200 in section 4.  Many of the other studies in section 4 are under continued study, and the side effects are still being evaluated.  We strongly agree with the reporting of side effects in clinical trials.

Comment 3: The most important statements regarding the side effects of the current therapy and the tolerability of the patients to the therapy should be added in the conclusion as well. This could be good starting point for further development of improved therapeutics given the limitations in efficiency of the therapeutic choices as well as the safety profile of the available therapy.

Response 3: We added a statement regarding the diminished quality of life of chemotherapy in lines 287-289.  We agree that novel therapeutics with better side effect profiles are needed.